# The Evolving Role of Cardiac Imaging in Hypertrophic Cardiomyopathy: Diagnosis, Prognosis, and Clinical Practice

**DOI:** 10.3390/biomedicines13092138

**Published:** 2025-09-01

**Authors:** Ilaria Dentamaro, Marco Maria Dicorato, Alessio Falagario, Sebastiano Cicco, Sergio Dentamaro, Michele Correale, Vincenzo Manuppelli, Gaetano Citarelli, Francesco Mangini, Corrado Fiore, Paolo Colonna, Enrica Petruccelli, Laura Piscitelli, Guido Giovannetti, Michele Davide Latorre, Cinzia Forleo, Paolo Basile, Maria Cristina Carella, Vincenzo Ezio Santobuono, Marco Matteo Ciccone, Andrea Igoren Guaricci

**Affiliations:** 1University Cardiology Unit, Interdisciplinary Department of Medicine, Polyclinic University Hospital, University of Bari “Aldo Moro”, 70121 Bari, Italy; m.dicorato20@studenti.uniba.it (M.M.D.); a.falagario2@studenti.uniba.it (A.F.); micheledavide.latorre@policlinico.ba.it (M.D.L.); cinzia.forleo@uniba.it (C.F.); paolo.basile@uniba.it (P.B.); m.carella31@phd.uniba.it (M.C.C.); eziosantobuono@gmail.com (V.E.S.); marcomatteo.ciccone@uniba.it (M.M.C.); andreaigoren.guaricci@uniba.it (A.I.G.); 2Internal Medicine Unit “Guido Baccelli”-Arterial Hypertension Unit “Anna Maria Pirrelli”, Department of Precision and Regenerative Medicine and Jonic Area (DiMePReJ), Polyclinic University Hospital, University of Bari “Aldo Moro”, 70124 Bari, Italy; sebacicco@gmail.com; 3Vascular Surgery, Polyclinic University Hospital, University of Bari “Aldo Moro”, 70121 Bari, Italy; sergio.dentamaro@gmail.com; 4Cardiology Unit, Cardio-Thoracic Department, University Policlinico Riuniti of Foggia, 71122 Foggia, Italy; michele.correale@libero.it (M.C.); manuppelli.dr@gmail.com (V.M.); 5Cardiology Division, San Paolo Hospital, 70132 Bari, Italy; gaetanocita@libero.it; 6Cardiology Division, Miulli Hospital, 70021 Acquaviva delle Fonti, Italy; francescomangini.78@libero.it; 7Cardiology Department and EchoLab, Città di Lecce Hospital-GVM, 73100 Lecce, Italy; cardiologo85@gmail.com; 8Cardiology Department, San Giacomo Hospital, 70043 Monopoli, Italy; colonna@tiscali.it (P.C.); petruccellienrica@gmail.com (E.P.); 9Cardiology Unit, A. Perrino Hospital, 72100 Brindisi, Italy; piscitellilaura92@gmail.com; 10Cardiology Department, IRCCS Maugeri Hospital, 70124 Bari, Italy; guidogiovannettijr@gmail.com

**Keywords:** hypertrophic cardiomyopathy, left ventricular outflow tract obstruction, echocardiography, cardiac magnetic resonance, multimodality imaging, athlete’s heart

## Abstract

Hypertrophic cardiomyopathy (HCM) is a cardiac disorder characterized by unexplained left ventricular hypertrophy and a clinical presentation that is heterogeneous, ranging from asymptomatic cases to sudden cardiac death (SCD). The condition’s complex pathophysiology encompasses myocyte disarray, fibrosis, and impaired cellular metabolism. Advancements in non-invasive cardiac imaging, notably echocardiography and cardiac magnetic resonance (CMR), have led to substantial progress in the domains of early diagnosis, phenotypic characterization, and risk stratification. Echocardiography is the preferred diagnostic modality, as it provides a comprehensive evaluation of ventricular hypertrophy patterns, left ventricular outflow tract (LVOT) obstruction, mitral valve abnormalities, left atrial size, and diastolic function. Novel techniques, such as speckle-tracking strain imaging, have emerged as means to detect subclinical myocardial dysfunction and to provide significant prognostic information. Cine-CMR sequences, tissue characterization with late gadolinium enhancement, and quantitative techniques such as strain imaging have been shown to enhance diagnostic precision and prognostic evaluation. The integration of multimodality imaging has been demonstrated to enhance the management of patients with HCM, both in the short term and in the long term, by facilitating individualized monitoring. This review summarizes the role of cardiac imaging in the comprehensive evaluation of HCM, emphasizing the impact of these methods on diagnosis, risk assessment, and personalized patient care, particularly in challenging clinical settings, such as cases of athlete’s heart and pathological ventricular hypertrophy.

## 1. Introduction

Hypertrophic cardiomyopathy (HCM) is a heart disease with a prevalence estimated between 1 in 200 and 1 in 500 people. It is characterized by an increased left ventricular (LV) wall thickness that cannot be attributed solely to abnormal loading conditions, such as hypertension or valvular heart disease [1]. HCM is characterized by a heterogeneous phenotype, as well as systolic and diastolic dysfunction and histopathological changes, including myocyte disarray and myocardial fibrosis. Its complex pathophysiology involves impaired biophysical properties of cardiomyocytes, disrupted calcium handling, and abnormal cellular metabolism [2,3]. Sarcomeric hypertrophic cardiomyopathy is an autosomal-dominant genetic heart disease, resulting from mutations in genes encoding sarcomeric proteins. These mutations increase myocyte energy demand, ultimately leading to cardiac hypertrophy. The mutations also trigger a cardiomyocyte replication stress response, contributing to pathological myocardial remodeling in sarcomeric cardiomyopathy [2,4,5]. The final phenotype is influenced by environmental, genetic, and epigenetic factors, resulting in variable disease expression. HCM presents with a range of phenotypes, including asymmetric septal hypertrophy with or without left ventricular outflow tract obstruction, moderate LV dilation with or without apical aneurysm formation, and, in some cases, severe end-stage dilation associated with refractory heart failure (HF). This phenotypic variability results from a combination of factors, including preload, afterload, wall stress, and myocardial ischemia due to microvascular dysfunction and thrombosis [4,6]. Patients with HCM also face an elevated risk of sudden cardiac death(SCD), with an estimated annual incidence of 0.59% to 1%. Myocardial fibrosis is a key contributor to this risk, though it is not the only factor involved [7]. Due to its variable clinical presentation and potentially life-threatening complications, an early and accurate diagnosis of HCM is crucial. Non-invasive imaging is a cornerstone of HCM diagnosis, with echocardiography and cardiac magnetic resonance imaging (CMR) as the principal modalities. Recent advances in these techniques allow for a more detailed assessment of myocardial architecture, function, and tissue composition. Moreover, traditional diagnostic workflows are prone to variability due to human-related factors such as operator dependency and interpretative bias, which underscores the growing importance of standardized and automated imaging approaches. This enables earlier detection, more precise phenotyping, and the development of targeted management strategies [8]. This review explores the role of cardiac imaging in HCM, focusing on its utility in diagnosis, risk stratification, and long-term follow-up.

## 2. Materials and Methods

This article is a structured narrative review that summarizes the current evidence on the role of cardiac imaging techniques in diagnosing, risk-stratifying, and managing HCM. A non-systematic literature search was conducted using the PubMed/MEDLINE and Google Scholar databases. Articles published from January 2015 to May 2025 were considered eligible. The time window was selected to reflect the period of major technological innovation in image processing and automation. The search strategy included combinations of the following terms: “hypertrophic cardiomyopathy,” “cardiac imaging,” “echocardiography,” “cardiac magnetic resonance,” “computed tomography,” “strain imaging,” “artificial intelligence,” and “risk stratification.” We included peer-reviewed, full-text articles written in English that encompassed original research studies, systematic reviews, meta-analyses, expert consensus documents, and clinical guidelines. We excluded case reports, conference abstracts, editorials, non-English publications, and non-peer-reviewed materials. Articles were selected based on their relevance to the topic and their contribution to understanding both established and emerging cardiac imaging modalities in HCM. Figure 1, Figure 2 and Figure 3 are representative clinical images obtained in our echocardiography and CMR laboratory. Echocardiographic images were acquired using commercially available ultrasound platforms (EPIQ 7, Philips Healthcare, Andover, MA, USA; Vivid E95, GE Healthcare, Chicago, IL, USA), following standard transthoracic protocols recommended by the ASE and EACVI. Strain analysis was performed with vendor-specific software. CMR images were obtained using a 1.5 T system (Magnetom Avanto, Siemens Healthineers, Erlangen, Germany) with cine SSFP, mapping sequences, and late gadolinium enhancement in line with international recommendations. These figures are provided for illustrative purposes only and do not constitute original research data. Written informed consent for publication was obtained from all patients. 

## 3. Overview

Over the past five decades, the epidemiology of HCM has undergone a significant transformation, partly due to advancements in imaging modalities. The widespread availability of echocardiography and CMR has enabled earlier detection of the disease and reduced the proportion of strictly familial cases [9]. Although historically considered a distinct hypertrophic phenotype, HCM shares key clinical and pathophysiological features with restrictive cardiomyopathy. Both conditions can present with diastolic dysfunction, though the timing and severity may differ. Echocardiography remains the primary imaging modality, providing a rapid, real-time evaluation of cardiac function, valvular performance, and diastolic filling. It excels at visualizing valve morphology and assessing diastolic function. It has a key function in the ongoing clinical surveillance of HCM patients. However, echocardiographic measurements of systolic function and ventricular volumes are subject to significant intra- and inter-operator variability, often leading to underestimation of LV volumes and overestimation of ejection fraction compared with CMR [10]. In recent years, noninvasive imaging has advanced substantially, particularly in the areas of CMR and cardiac computed tomography (CT). These imaging techniques combine morphological, functional, and tissue-specific information, enabling earlier and more precise diagnoses. The quantification of LV and right ventricle (RV) volumes and function is most reliably achieved through CMR, which is regarded as the gold standard imaging technique. Furthermore, CMR enables in-depth tissue characterization, which is essential for differentiating between various forms of non-ischemic cardiomyopathy [11,12]. While cardiac CT has traditionally been used to assess coronary arteries, contemporary protocols also allow for evaluating ventricular volumes, performing functional analyses, and, in select cases, characterizing tissue [13]. Integrating multiple imaging modalities—enhanced, when applicable, by artificial intelligence-driven analytics—allows clinicians to streamline diagnostic and therapeutic decision-making, improving risk stratification, particularly in relation to arrhythmic events and HF progression [14,15,16,17]. A comparative summary of conventional and emerging cardiac imaging modalities in HCM is provided in Table 1, highlighting their respective strengths, limitations, and roles in clinical practice.

## 4. Echocardiography

Echocardiography plays a pivotal role in diagnosing, risk-stratifying, managing, and following up with patients with HCM. A definitive diagnosis requires a systematic and comprehensive echocardiographic evaluation, including confirmation of LV hypertrophy and assessment of LVOT obstruction. Key components of the examination include evaluating systolic anterior motion (SAM) of the mitral valve, analyzing both systolic and diastolic LV function in detail, and measuring left atrial (LA) size. In recent years, advanced echocardiographic techniques have significantly enhanced the ability to detect subclinical myocardial dysfunction, offering deeper insights into early disease processes [18]. Transthoracic echocardiography is also strongly recommended for screening first-degree relatives of individuals with HCM, given its noninvasive nature and diagnostic utility.

### 4.1. Confirming the Presence of Left Ventricular Hypertrophy

In HCM, LVH most commonly affects the basal interventricular septum, though it can also extend to other regions such as the lateral wall, posterior septum, or apex. The presence of asymmetric septal hypertrophy (ASH) remains a defining characteristic. Diagnostic criteria include a maximum wall thickness > 15 mm in any segment and/or a septal-to-posterior wall thickness ratio > 1.3 in normotensive individuals or >1.5 in hypertensive patients [19]. In the pediatric population, z-scores adjusted for body size are used, with thresholds > 2.5 in asymptomatic children without family history, or >2.0 in those with positive family history or genetic testing. Hypertrophy may be focal despite normal overall LV mass. In approximately 75% of patients, the classic form of HCM is characterized by a hypertrophied but non-dilated left ventricle with preserved or even elevated systolic function (EF > 65%). This phenotype usually manifests in adolescence and reaches a steady state in early adulthood [8]. Massive hypertrophy (>30 mm) is associated with adverse prognosis. Distinct variants include mid-ventricular HCM, linked to arrhythmias, necrosis, and apical aneurysms, and apical HCM, defined by apical thickening and a “spade-like” LV cavity (5–25% of cases) [20,21]. Genotype-positive individuals or those with a family history are considered at risk even in the absence of hypertrophy, and may still face SCD risk despite normal wall thickness [19]. Comprehensive assessment requires measuring wall thickness from base to apex across all segments. Importantly, asymmetric LVH is not specific to HCM and may also occur in other conditions such as systemic or RV hypertension, aortic stenosis, septal tumors, Fabry disease, Friedreich’s ataxia, and various storage disorders [22,23].

### 4.2. Assessment of Latent Left Ventricular Outflow Tract Obstruction

It is essential to differentiate obstructive from non-obstructive HCM, as therapeutic decisions depend on the presence of LVOT obstruction (LVOTO). Management is guided by the peak instantaneous gradient, not the mean gradient [24]. Assessment should include 2D and Doppler echocardiography with physiological provocation maneuvers performed in the sitting, semi-supine, and standing positions (if no gradient is initially observed), including a Valsalva maneuver. About one-third of patients with HCM demonstrate LVOTO at rest, characterized by a peak pressure gradient of 30 mmHg or higher. Another third demonstrate latent or labile obstruction, with resting gradients of less than 30 mmHg but rising to at least 30 mmHg upon physiological provocation. The remaining third have a non-obstructive phenotype, characterized by gradients of less than 30 mmHg at rest and with provocation [25]. For symptomatic patients without a significant bedside gradient (≥50 mmHg), exercise stress echocardiography is recommended. Pharmacologic stress with dobutamine is discouraged due to its nonphysiological nature and poor tolerability. Nitrates are unreliable and should be reserved for patients unable to undergo physiological testing [24].

### 4.3. Systolic Anterior Motion and Mitral Valve Abnormalities

SAM is a hallmark of HCM, typically detected by M-mode echocardiography and defined by anterior mitral leaflet or chordal contact with the interventricular septum, often associated with mid-systolic notching of the aortic valve. In obstructive HCM, septal hypertrophy and a reduced LV cavity contribute to dynamic LVOTO, frequently accompanied by posteriorly directed mitral regurgitation and mitral annular calcification. The severity of LVOTO is dynamic and influenced by preload, afterload, and contractility. Recent evidence highlights the role of abnormal mitral valve positioning, whereby systolic flow drags the anterior leaflet toward the outflow tract, worsening obstruction [26,27,28]. When an LVOT gradient is detected, alternative causes such as subaortic membranes or mid-cavity obstruction should be ruled out [26]. Beyond myocardial involvement, structural mitral valve abnormalities are common in HCM and contribute to both SAM and LVOTO. Transthoracic echocardiography is pivotal for assessing mitral valve anatomy, quantifying SAM, and evaluating mitral regurgitation and related abnormalities. Frequent findings include elongation and increased laxity of the anterior mitral leaflet, which is significantly longer in HCM patients compared to controls. Fibrotic and thickened secondary chordae may tether the anterior leaflet toward the septum, exacerbating SAM [29,30,31]. Papillary muscle abnormalities—including anterior/basal displacement or direct insertion into the anterior leaflet—also promote LVOTO by reducing leaflet-septum distance. Displacement of the papillary muscles alone is sufficient to cause SAM and LVOTO [32]. Additionally, anatomical variations like bifid anterior papillary muscles or direct insertion of papillary muscles exacerbate the degree of obstruction

### 4.4. Left Atrial Enlargement

LA volume is predominantly influenced by diastolic dysfunction, mitral regurgitation, and atrial myopathy. Although LA enlargement assessed by linear dimensions has been shown to independently predict long-term prognosis in patients with HCM, it is important to note that linear measurements may underestimate the true LA size due to the chamber’s often asymmetric remodeling. Current guidelines from the American Society of Echocardiography (ASE) and the European Association of Cardiovascular Imaging (EACVI) recommend quantifying LA size by indexing LA volume, which is calculated using the biplane area-length method or the method of disks, to body surface area. The normal indexed LA volume is reported as 22 ± 6 mL/m^2^ [33]. Furthermore, an indexed LA volume exceeding 34 mL/m^2^ has been identified as an independent prognostic marker that correlates with greater LV hypertrophy (Figure 1), more severe diastolic dysfunction, atrial fibrillation onset, and worse cardiovascular outcomes [34,35].

**Figure 1 biomedicines-13-02138-f001:**
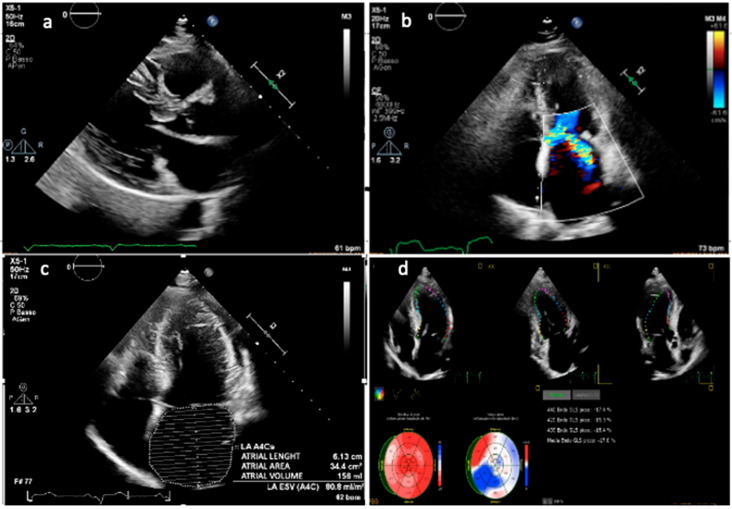
Transthoracic echocardiographic evaluation in a patient with hypertrophic obstructive cardiomyopathy. (**a**) Parasternal long-axis view showing asymmetric septal hypertrophy with systolic anterior motion of the mitral valve and elongation of leaflets. (**b**) Apical four-chamber view with color Doppler demonstrating turbulent flow across the left ventricular outflow tract and significant mitral valve regurgitation due to SAM. (**c**) Apical four-chamber view highlighting marked left atrial dilation. (**d**) Speckle-tracking echocardiography showing global longitudinal strain analysis with reduced strain values in the hypertrophied septal segments. Representative case from a single patient with hypertrophic obstructive cardiomyopathy.

### 4.5. Diastolic Function

Diastolic dysfunction is a key pathophysiological feature of HCM, resulting from a combination of increased LV wall thickness, impaired relaxation, reduced compliance, myocardial ischemia, and fibrosis [36,37]. A restrictive LV filling pattern is associated with worse prognosis [38]. While early evaluation relied on invasive measures such as pulmonary capillary wedge pressure and LV end-diastolic pressure (LVEDP), Doppler echocardiography has become the standard noninvasive tool. However, its accuracy may be limited by factors including heart rate, age, and loading conditions. Standard Doppler parameters—such as E-wave velocity, E/A ratio, and deceleration time—do not consistently correlate with invasively measured LVEDP in HCM. Among noninvasive indices, the E/e′ ratio derived from tissue Doppler imaging (TDI) of the mitral annulus is considered a better estimate of LV filling pressures, though with variable predictive accuracy across studies [39]. Therefore, a comprehensive multiparametric approach is recommended, integrating transmitral inflow, TDI, pulmonary venous flow, left atrial (LA) size and volume, and tricuspid regurgitation (TR) velocity [38]. According to ASE/EACVI guidelines, severe diastolic dysfunction is diagnosed when more than half of the following are present: E/e′ >14, LA volume index > 34 mL/m^2^, pulmonary vein atrial reversal duration ≥ 30 ms, and TR peak velocity > 2.8 m/s. Importantly, diastolic impairment may precede LV hypertrophy in genotype-positive individuals. In such cases, early abnormalities in diastolic filling and mitral annular velocities can be detected by echocardiography, suggesting that myocardial dysfunction may occur independently of structural remodeling.

### 4.6. Strain Imaging

Recently, myocardial strain imaging by speckle-tracking echocardiography (STE) has emerged as a valuable HCM evaluation tool. Unlike conventional echocardiographic parameters, strain analysis provides a sensitive quantification of myocardial deformation. This enables the early detection of subtle systolic dysfunction, even when left ventricular ejection fraction remains preserved. Multiple studies have demonstrated that global longitudinal strain (GLS) is significantly reduced in HCM patients compared to healthy controls. GLS correlates with the extent of myocardial fibrosis detected by cardiac magnetic resonance imaging with late gadolinium enhancement (LGE) [40]. Furthermore, segmental strain patterns can distinguish between different HCM phenotypes, such as asymmetric septal hypertrophy and apical HCM, and predict regions at risk for arrhythmogenesis [41]. Beyond its diagnostic utility, GLS has shown independent prognostic value in HCM (Figure 1) [42,43].

## 5. Cardiac Magnetic Resonance

CMR imaging plays an increasingly important role in evaluating patients with HCM due to its ability to provide useful information for diagnosis and risk stratification [44]. CMR imaging enables precise measurement of left ventricular wall thickness, including both regional hypertrophy and aneurysmal segments [45]. Cine-MR sequences offer detailed visualization of myocardial and valvular motion, enabling qualitative assessment of abnormal flow patterns. Integrated with phase-contrast sequences, they permit quantitative flow analysis, including the capability to quantify mitral regurgitation. Furthermore, tissue characterization, including detecting fibrosis via LGE and quantifying it, provides critical insights into the severity and distribution of myocardial scarring [46].

### 5.1. Cine-MR and Phase-Contrast Velocity Mapping Sequences

CMR remains the gold standard method for detailed morphological and functional assessment of HCM. Compared with echocardiography, its superior signal-to-noise ratio (SNR) allows for a more precise assessment of wall thickness and ventricular cavity dimensions, particularly in technically challenging regions, such as the apical and lateral walls. This enhanced resolution enables identification of subtle or “occult” hypertrophy, which may have clinical significance despite being mildly severe. The primary imaging characteristic of HCM is LV wall thickening exceeding 15 mm in the absence of other causes [1], although thinning may occur in other segments or during the final stages of the disease [47]. There are multiple HCM variants, including the apical phenotype, in which myocardial thickness may not exceed 15 mm, and mass-like HCM, in which the wall thickening is often found in the anteroseptal and anterior walls. Other imaging findings that can support the diagnosis of HCM include prominent myocardial crypts and altered papillary muscle and trabecular geometry [48]. There is often a “hyperdynamic” LV contraction pattern with near-complete systolic emptying of the mid-level and apical LV cavity, especially in the apical hypertrophy variant. A notable strength of cine-MR is its ability to detect apical aneurysms, particularly those associated with mid-ventricular or purely apical HCM variants (Figure 2). These findings may be overlooked in standard echocardiographic examinations. Cine-MR enables the identification of ventricular obstruction, which predominantly, but not exclusively, occurs in the LVOT. This obstruction can be seen through the characteristic “dephasing artifact,” which appears as a signal void on gradient-echo sequences. This void is caused by turbulent, high-velocity flow in the narrowed outflow tract. While this artifact is readily recognizable and thus facilitates a straightforward visual assessment of obstruction [49,50], quantitative analysis is considerably more challenging. The anatomic geometry of the LVOT is difficult to delineate precisely across multiple imaging planes. Dynamic maneuvers designed to provoke or exacerbate obstruction cannot typically be performed within the MRI scanner. This restricts accurate quantification under physiologically stressed conditions. Furthermore, cine-MR offers a detailed evaluation of mitral valve behavior, including SAM [49,50]. SAM is frequently associated with varying degrees of mitral regurgitation beyond contributing to LVOT obstruction. Although velocity-acceleration measurements related to SAM are generally more reliable via echocardiography than CMR, CMR is essential for quantifying regurgitant volumes and fractions, especially in cases involving eccentric jets. This is achieved by combining phase-contrast velocity mapping of forward flow in the aorta or pulmonary artery with stroke volume measurements from cine MRI [51]. This integrative approach provides a thorough evaluation of mitral regurgitation severity and assists in optimizing clinical management. One application of cine-MR is CMR feature tracking. This technique can detect early LV and left atrial (LA) dysfunction in HCM patients, even when LV ejection fraction (EF) is preserved [52]. Specifically, reduced strain parameters correlate with elevated biomarkers, such as troponin T and NT-proBNP, and predict adverse outcomes, particularly when the peak diastolic strain rate (PDSR) is reduced [53]. Abnormal strain values, such as reduced global circumferential and radial strain, are associated with LVOTO and arrhythmic risk [54]. Strain alterations often correspond to fibrotic areas seen on LGE; segmental longitudinal strain aids in fibrosis detection [55]. Further extensions of cine-MR, such as CMR tagging and 3D strain, provide a detailed assessment of regional function. These techniques can differentiate focal HCM from masses and demonstrate functional improvement after septal ablation, which supports reverse remodeling.

**Figure 2 biomedicines-13-02138-f002:**
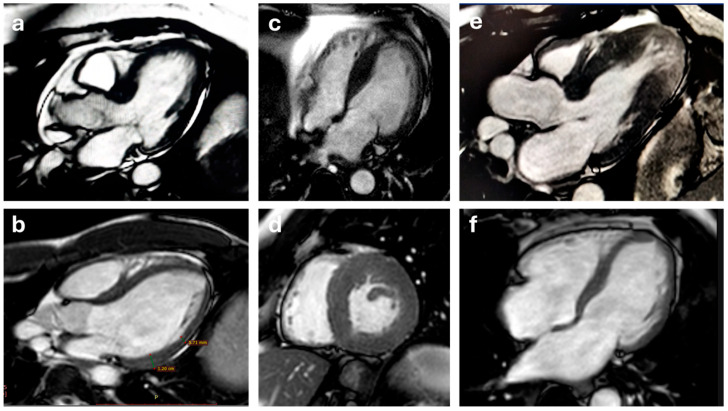
Different examples of hypertrophic cardiomyopathy variants displayed on cine-MRI sequences: basal septal distribution (**a**), basal posterior (**b**), mid-septal (**c**), concentric/uniform (**d**), diffuse apical (**e**), and focal apical (**f**). Examples obtained from different patients, illustrating the heterogeneity of HCM phenotypes.

### 5.2. T1 and T2 Mapping

T1 mapping techniques provide a quantitative assessment of diffuse myocardial fibrosis by measuring T1 relaxation times both before (native T1) and after (post-contrast T1) gadolinium-based contrast administration. An increased native T1 value typically reflects expansion of the interstitial space, indicative of diffuse fibrosis, and is often corroborated by an elevated extracellular volume (ECV) fraction. Unlike raw T1 values, ECV calculation is field-strength independent, making it a particularly robust parameter [56]. The most commonly employed method for T1 mapping is the Modified Look-Locker Inversion recovery (MOLLI) sequence or its shortened variant (shMOLLI), which uses radiofrequency pulses and signal acquisitions to measure T1 values. It can be combined with motion-correction during a single breath hold. ECV is then determined by comparing pre- and post-contrast T1 values from both myocardium and blood, factoring in hematocrit. When hematocrit is not directly available, a “synthetic” hematocrit can be estimated from the longitudinal relaxation rate of blood, enabling the calculation of synthetic ECV (ECVsyn) [45]. Native T1 mapping can detect subtle fibrotic changes in areas not evident on LGE [57], thereby highlighting early structural alterations within the myocardium. By quantifying interstitial fibrosis, T1 mapping offers critical insights into disease progression and risk stratification, thus enabling more individualized patient management. In HCM, native T1 values frequently manifest as prolonged, correlating with the extent of ventricular wall thickening. This association suggests that native T1 may serve as an important marker of disease severity [58]. Furthermore, patients with HCM often demonstrate reduced post-contrast T1 values, indicating the presence of diffuse interstitial fibrosis outside the regions identified by LGE. ECV tends to occupy the higher spectrum of the normal range, even in myocardial segments that do not exhibit LGE [59]. As myocardial wall thickness increases, both native T1 and ECV tend to elevate correspondingly, distinguishing HCM from hypertensive heart disease, which typically displays either normal or slightly lower global T1/ECV values. Moreover, T1 mapping provides differential diagnosis across various etiologies. For instance, cardiac amyloidosis is characterized by markedly prolonged native T1 and markedly increased ECV, reflecting extensive amyloid protein deposition [23,60]. This distinct pattern contrasts sharply with other causes of left ventricular hypertrophy (LVH), including HCM or hypertensive heart disease. Fabry disease is characterized by reduced native T1 values, which result from glycolipid accumulation within cardiac myocytes T1 mapping thus becomes a particularly useful modality in differentiating Fabry disease from infiltrative or fibrotic processes. Hypertrophic Cardiomyopathy (HCM) and Hypertensive Heart Disease: These conditions typically present with moderately elevated or near-normal T1 values and ECV levels [59]. Although these parameters may be somewhat increased, they generally remain below those observed in cardiac amyloidosis [23]. By integrating T1 mapping and ECV quantification, clinicians can elucidate both the nature of myocardial involvement and the degree of extracellular matrix expansion. This comprehensive tissue characterization not only facilitates more accurate differentiation among diverse cardiomyopathies—particularly in cases where standard imaging modalities yield inconclusive findings—but also informs individualized clinical management, ranging from determining the appropriate timing for therapeutic interventions to guiding long-term follow-up and prognostication [45,59]. T2-weighted CMR imaging has traditionally been used to detect myocardial edema by exploiting the increase in T2 relaxation times associated with inflammation. Nevertheless, traditional T2-weighted imaging techniques frequently suffer from limited reproducibility and rely heavily on subjective interpretation of the images. T2 mapping mitigates these challenges by directly measuring local T2 values, typically through the acquisition of multiple images at different echo times (e.g., 0, 25, and 50 ms) and fitting the data to a T2 decay curve [59]. This approach provides a quantitative assessment of tissue edema, though it remains sensitive to factors such as T1 weighting and off-resonance effects. In the context of HCM, recent investigations employing T2 mapping at 3 T have revealed that tissue remodeling can occur even in myocardial segments that do not exhibit overt hypertrophy. Huang et al. found that native T1 and T2 values were significantly elevated in both non-hypertrophic and hypertrophic segments of HCM patients compared to controls [61]. The percentage of wall thickening was preserved in non-hypertrophic segments, while significantly impaired in hypertrophic segments. The native T1 values of severely hypertrophied segments in HCM were significantly elevated compared to those of segments with mild or moderate hypertrophy. In more markedly hypertrophic segments, T2 values are further elevated, in parallel with reductions in systolic wall thickening [61]. These findings point to the utility of T2 mapping as an adjunct to standard morphological measures, allowing for earlier identification of pathological changes and potentially refining both diagnostic precision and clinical management of HCM.

### 5.3. Late Gadolinium Enhancement

Gadolinium-based (GbA) contrast agents accumulate within the extracellular space and consequently shorten the T1 relaxation time in surrounding tissues. After a typical post-injection delay of about 10–20 min, the signal arising from normal myocardium is “nulled,” enabling areas of GbA accumulation to appear hyperintense. This phenomenon underpins the clinically established role of LGE in distinguishing different scar etiologies, as each disease process exhibits a characteristic LGE distribution pattern. LGE imaging is performed using inversion-recovery sequences, where the inversion time is carefully set to null the signal from healthy myocardium, thereby maximizing contrast between normal myocardium, blood, and scar tissue [62]. The standard approach employs T1-weighted gradient echo sequences, with either a manually adjusted IT or a phase-sensitive inversion recovery (PSIR) technique. PSIR consists of acquiring the phase information of the CMR signal, which allows for the distinction between positive and negative magnetization values. This contrasts with magnitude-based LGE, which only uses the signal’s magnitude and can be less sensitive to subtle changes in tissue properties. Optimal IT selection often involves a preliminary Look-Locker sequence prior to LGE scanning. The development of PSIR sequences has notably minimized issues arising from inaccurate inversion time adjustments [63], as a dedicated phase map is acquired in tandem with the LGE images, correcting for polarity misregistration but requiring a longer scan duration [58,64]. In alignment with current guidelines [56], LGE imaging typically includes a comprehensive short-axis stack covering the entire LV, as well as the three-standard long-axis views. Over 50% of patients with HCM exhibit some degree of fibrosis, most frequently in a “patchy” pattern localized to regions of marked hypertrophy, though subendocardial or mid-wall patterns can also be observed (Figure 3). Notably, fibrotic involvement of the papillary muscles is not uncommon [46,65]. Beyond its diagnostic utility, LGE also holds prognostic significance: an increased extent of fibrosis correlates strongly with the risk of ventricular arrhythmias [66]. In a 2014 study, Chan et al. found that, compared with patients without LGE, the risk of SCD increased substantially across the range of LGE extent, with LGE ≥15% of LV mass conferring more than a twofold increase in risk, even in patients otherwise considered low risk. This concept has been incorporated into the most recent guidelines [67]. In fact, when LGE exceeds 15% of LV mass, patients face a significantly heightened risk of life-threatening arrhythmias, supporting more proactive strategies such as prophylactic implantable cardioverter implantation in selected cases [68]. However, debate remains over the optimal method of LGE quantification, notably whether to employ a threshold of five or six standard deviations from a remote reference region or utilize a more manually derived regional cutoff.

**Figure 3 biomedicines-13-02138-f003:**
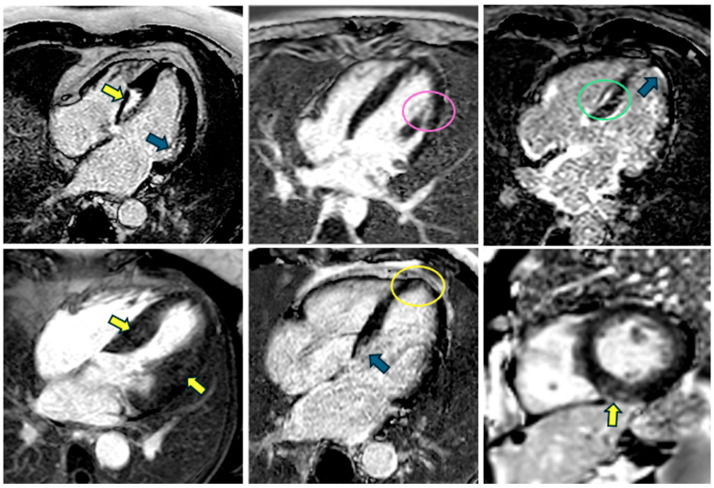
Different patterns of late gadolinium enhancement that can be seen in patients with hypertrophic cardiomyopathy: patchy (yellow arrows and yellow circle), subendocardial (blue arrows), mid-wall (green circle), and involvement of the papillary muscles (pink circle). Examples obtained from different patients, showing various LGE distribution patterns.

### 5.4. Perfusion

CMR perfusion imaging is capable of detecting myocardial perfusion abnormalities in patients with HCM, which are believed to reflect underlying microvascular dysfunction—a common feature in HCM that is often associated with myocardial hypertrophy and the presence of LGE [67]. First-pass perfusion CMR frequently reveals resting perfusion abnormalities in a significant number of HCM patients, which tend to correlate with the extent and location of myocardial fibrosis and the degree of hypertrophy [69]. Adenosine stress CMR detected perfusion defects in more than 40% of HCM patients, which were associated with higher indexed LV mass, apical aneurysms, and increased prevalence of NSVT [70]. Camaioni et al., in a study involving 101 HCM patients with non-obstructive coronary arteries, quantitative stress perfusion CMR showed that both global myocardial blood flow (MBF) and myocardial perfusion reserve (MPR) were significantly lower in HCM patients compared to controls. These reductions in perfusion metrics were more pronounced with increasing left ventricular mass, maximal wall thickness, and LGE. Importantly, even myocardial segments without hypertrophy or fibrosis demonstrated impaired stress MBF and MPR, suggesting that perfusion abnormalities may occur early in the disease process [71]. Notably, rest perfusion abnormalities identify patients with an increased incidence of episodes of NSVT, independently from the presence of myocardial fibrosis [67].

### 5.5. Evaluation of Intraventricular Flow Dynamics in HCM

Recently, assessing intraventricular flow dynamics has become a popular way to evaluate cardiac function beyond traditional structural and functional parameters. Techniques such as blood speckle imaging (BSI) and vector flow mapping (VFM) allow for a more detailed, physiologically insightful analysis of intraventricular blood flow patterns. These methods enable visualization of flow vectors, vortices, and regions of energy dissipation in real time. These methods provide novel information on the efficiency of ventricular filling and ejection, which is particularly relevant in diseases characterized by complex flow alterations, such as HCM. A 2019 study by Cao et al. [72] using VFM in patients with non-obstructive HCM demonstrated that diastolic energy loss (EL) in the LV is significantly reduced compared to healthy controls despite preserved ejection fraction. Interestingly, systolic EL tended to increase, possibly reflecting compensatory mechanisms. Furthermore, left atrial EL was significantly lower across all three functional phases (reservoir, conduit, and contraction), suggesting early atrial dysfunction. These findings correlated with impaired atrial strain parameters, such as reduced peak atrial longitudinal strain and increased atrial stiffness. This indicates that both ventricular and atrial subclinical dysfunction can be detected through this energetic approach [39]. Most studies on intraventricular flow dynamics and energy loss have been conducted using echocardiography due to its wide availability, high temporal resolution, and feasibility in clinical practice. However, CMR with 4D flow imaging technology allows better visualization and quantification of blood flow in three spatial dimensions over time, providing a comprehensive view of hemodynamics. The ability to noninvasively quantify intracardiac flow and energetic parameters using 4D flow CMR could significantly advance the comprehensive assessment of flow mechanics in HCM and other cardiomyopathies [73].

## 6. Computed Tomography

CMR is the preferred method for morphological and tissue characterization in HCM. However, cardiac computed tomography (CCT) has gained recognition as a valuable complementary imaging tool, especially for patients for whom CMR is contraindicated, such as those with implanted cardiac devices or severe claustrophobia. CCT provides high-resolution anatomical detail and is particularly effective in quantifying myocardial thickness and identifying structural contributors to LVOT obstruction, such as abnormal septal morphology or papillary muscle displacement. CCT can also visualize the relationship between the hypertrophied septum and the mitral valve. The latest generation of CT scanners, including dual-source systems and wide-detector technologies, can quantify biventricular volumes and systolic function. These advances allow full cardiac volume acquisition within a single heartbeat. Furthermore, reduced gantry rotation time enhances temporal resolution, enabling more accurate identification of the end-systolic and end-diastolic phases [67,74]. Beyond morphological evaluation, CCT allows for simultaneous assessment of coronary anatomy, which is essential for patients with HCM presenting with chest pain or coexisting cardiovascular risk factors [67]. Emerging techniques, such as dual-energy cardiac CT, enable tissue characterization, including the assessment of late iodine enhancement (analogous to LGE) and ECV. These techniques potentially mirror what is obtained with CMR. Though promising, these applications require further validation before routine use [67,75]. Despite these advancements, limitations remain. CCT involves exposure to ionizing radiation and iodinated contrast agents. Optimal image quality requires a low heart rate and regular rhythm. Additionally, vasodilators, which are used to enhance image acquisition, may be contraindicated in patients with severe LVOT obstruction. Therefore, for patients not suited for CCT, alternative imaging modalities, such as CMR perfusion or PET, may be preferable for assessing coronary function. However, larger prospective studies are needed before this technique can be routinely used in clinical practice.

## 7. Recent Advances and Emerging Techniques

### 7.1. Advanced Myocardial Strain Imaging

Myocardial strain imaging has emerged as a sensitive and clinically relevant tool in the evaluation of patients with HCM. In HCM, GLS is consistently reduced, particularly in hypertrophied segments, and correlates well with LGE on CMR, suggesting its potential as a surrogate for myocardial fibrosis. Recent studies also highlight that segmental strain patterns may differ based on genotype, helping distinguish sarcomeric from non-sarcomeric disease. Beyond 2D GLS, novel modalities such as three-dimensional (3D) strain imaging have been increasingly investigated. 3D STE offers improved spatial resolution and comprehensive assessment of myocardial deformation, including longitudinal, circumferential, and radial components. This allows for more accurate characterization of complex deformation patterns, particularly in asymmetric or apical HCM. For instance, Kim et al. demonstrated that 3D strain identified early mechanical dysfunction in genotype-positive patients with no hypertrophy, suggesting its role in detecting subclinical disease. LA strain has also gained attention as a marker of diastolic burden and atrial myopathy. In HCM, impaired LA reservoir and conduit strain have been associated with atrial fibrillation, elevated LV filling pressures, and worse outcomes. LA strain provides superior predictive value compared to LA volume alone and may be particularly useful in genotype-positive/phenotype-negative individuals or early phenotypes. This has been supported by several recent studies, including a multicenter prospective trial by Bytyçi et al. [76], which found LA stiffness index to be an independent predictor of cardiovascular events in HCM. Similarly, right ventricular (RV) strain has emerged as an important, yet often underrecognized, parameter in HCM. Though traditionally viewed as a left-sided disease, RV dysfunction has been reported in a subset of patients and correlates with symptoms, exercise capacity, and pulmonary pressures. Feature-tracking CMR and STE have both demonstrated reduced RV longitudinal strain in HCM patients with preserved LVEF. In a study by Negri et al. [76], impaired RV strain was independently associated with reduced functional capacity and atrial arrhythmias. Feature-tracking CMR (CMR-FT) further complements echocardiographic findings, allowing accurate deformation assessment from standard cine images. It provides robust and reproducible quantification of LV, LA, and RV strain without the limitations of acoustic windows [77]. When integrated with T1 mapping and LGE, CMR-FT may enhance phenotyping and improve risk stratification.

### 7.2. Artificial Intelligence in Cardiac Imaging for HCM 

The integration of artificial intelligence (AI) into cardiac imaging workflows has the potential to revolutionize the diagnosis, phenotyping, and risk stratification of HCM. Leveraging machine learning (ML) and deep learning (DL) techniques, AI enhances image acquisition, interpretation, and post-processing across various modalities, including echocardiography, CMR, and CT. In echocardiography, AI-based platforms have demonstrated utility in automating key measurements such as left ventricular wall thickness and GLS, significantly reducing intra- and inter-observer variability. Recent developments in deep learning algorithms allow real-time interpretation and even automated classification of HCM phenotypes by analyzing large datasets of echocardiographic videos [78,79]. Furthermore, AI-assisted GLS quantification improves reproducibility and shortens analysis time, facilitating integration into routine clinical workflows [80]. CMR has also benefited significantly from AI-enhanced image processing. Deep learning tools can perform automated segmentation of the myocardium and identify areas of LGE, which is critical for fibrosis quantification and risk stratification. For example, Zhang et al. demonstrated that AI-based fibrosis quantification from LGE images was more predictive of adverse outcomes than manual segmentation, showing promise in improving arrhythmic risk prediction [81]. Moreover, AI models can analyze tissue characterization maps (T1/T2 mapping) to detect diffuse fibrosis, even when visual assessment is inconclusive [82]. AI is also advancing the field of 4D flow CMR by enabling rapid and automated assessment of intraventricular flow patterns, vorticity, and kinetic energy dissipation. These hemodynamic parameters may offer additional insights into diastolic function and energy inefficiency in HCM patients, which are difficult to assess using conventional techniques [83]. In cardiac CT, AI applications include coronary artery segmentation, plaque characterization, and myocardial thickness measurement. Although its role in HCM is still emerging, the potential for AI to combine anatomical and functional CT data could enhance pre-interventional planning, especially in patients with contraindications to CMR [16]. Looking forward, the integration of AI with multimodal data—including clinical variables, genomics, and imaging biomarkers—could support comprehensive predictive models for HCM outcomes. AI-enabled decision support systems may assist clinicians in identifying high-risk patients, tailoring surveillance, and guiding therapy in a standardized, reproducible manner [84]. Despite the promise, these applications require rigorous external validation and careful implementation to avoid overfitting and ensure generalizability. Regulatory, ethical, and interpretability considerations must also be addressed before AI can be fully embedded in clinical cardiology practice.

### 7.3. Emerging Molecular Imaging Techniques

While not currently routine in clinical evaluation of HCM, molecular imaging holds promise in providing pathophysiological insights beyond morphology and function. PET imaging with [18F]-FDG may help identify inflammatory activity in ambiguous cases or in patients with suspected overlap with myocarditis or sarcoidosis.

More recently, novel tracers targeting myocardial fibrosis such as 68Ga-FAPI have been explored in HCM and other cardiomyopathies. These allow direct visualization of fibroblast activation, potentially enabling early detection of fibrotic remodeling.

Müller et al. showed feasibility of fibrosis imaging in non-ischemic cardiomyopathies including HCM [85]. These techniques may complement LGE and mapping by offering molecular-level assessment of disease activity and could aid in future treatment monitoring or phenotypic differentiation.

## 8. The Differential Diagnosis from Athlete’s Heart: An Ongoing Challenge

Traditionally, HCM has been considered the leading cause of SCD with LVOTO and complex arrhythmias, particularly during high-intensity physical activity. However, more recent studies have challenged this long-standing perspective [86,87,88]. In a large, contemporary HCM registry, no increased incidence of SCD or life-threatening arrhythmias was observed among patients engaging in vigorous physical activity compared to those participating in moderate or no exercise [89]. This observation is especially relevant for genotype-positive, phenotype-negative individuals [90]. Sports eligibility for athletes with HCM has evolved significantly, driven by the development of more refined cardiovascular risk stratification tools and improved prognostic markers. Early recommendations by major cardiology societies [24,91,92] were conservative, generally restricting athletes with HCM from participating in high-intensity competitive sports. However, advances in cardiac imaging and a more nuanced understanding of individual risk have led to a shift toward personalized decision-making. According to the most recent American and European guidelines [1,20,93], competitive sports participation may be considered in carefully selected, low-risk individuals with HCM (Figure 4), while still acknowledging the unpredictable nature of exercise-related SCD in this population [1]. Comprehensive evaluation, including transthoracic echocardiography, CMR, and arrhythmia surveillance, is essential for risk assessment. Exclusion criteria for competitive sport participation include severe myocardial hypertrophy, presence of cardiac symptoms, resting or provoked LVOTO, and an HCM Risk-SCD score exceeding 4% [1]. Additional imaging findings associated with increased SCD risk include left atrial dilation, LVEF < 50%, moderate to severe mitral regurgitation, apical aneurysm, and extensive LGE on CMR involving >15% of myocardial mass [92,94]. Conversely, low- to moderate-intensity exercise is generally encouraged in all patients with HCM who are able to participate, as it has been shown to improve cardiorespiratory fitness, psychosocial well-being, and overall quality of life [1,95]. However, HCM Risk score has demonstrated some limitations too. Among the others, it has low sensitivity when the risk percentage increases [96]. Furthermore, in low-risk patients (score < 4%), it lacks sensitivity even in predicting long-term mortality. According to a recent study, advanced age, cerebrovascular events, and high neutrophil counts independently predicted mortality and have been proposed by the authors to implement the efficiency of the risk score [97]. These findings highlight the need for a patient-tailored assessment. Sports eligibility decisions should be individualized and re-evaluated at least annually. In this context, imaging parameters—particularly those incorporated into the HCM Risk-SCD score—remain fundamental, given their strong correlation with adverse outcomes, including SCD [92].

Imaging plays a pivotal role in distinguishing HCM from physiological adaptations observed in the athlete’s heart. Exercise-induced cardiac remodelling is a benign and adaptive process that can mimic pathological hypertrophy, particularly in highly trained individuals. The degree and distribution of left ventricular hypertrophy in athletes are affected by various factors such as the sport discipline, body composition, gender, age, ethnicity, as well as the intensity and length of training. Transthoracic echocardiography remains the first-line imaging modality, offering critical insights into cardiac structure and function. In athletes, isolated LV wall thickening without concurrent chamber dilation is rare. Wall thickness greater than 13 mm is uncommon, and values exceeding 15 mm are considered highly suggestive of pathology, particularly HCM. Moreover, hypertrophy in athletes tends to be symmetrically distributed as a physiological response to increased preload and afterload, while HCM more commonly exhibits asymmetric patterns [98,99]. LV wall thickness between 13 and 15 mm represents a diagnostic “grey zone,” where differentiation from mild HCM becomes particularly challenging. Echocardiographic features favouring a diagnosis of HCM include SAM of the mitral valve and the presence of a resting or provocable LVOT gradient. Advanced techniques such as speckle-tracking echocardiography have further improved diagnostic precision, with studies consistently demonstrating significantly lower GLS in HCM patients compared to athletes [100,101]. Additionally, HCM is associated with impaired systolic deformation and delayed diastolic untwisting, whereas athletes typically exhibit preserved or enhanced untwisting dynamics during early diastole. These strain-based parameters may enhance diagnostic accuracy in equivocal cases, although further validation is necessary before widespread clinical implementation. CMR imaging provides superior spatial resolution and detailed myocardial tissue characterization, making it an invaluable tool in differentiating HCM from athlete’s heart. The presence of LGE, indicative of myocardial fibrosis, is a hallmark of HCM and is generally absent in physiological hypertrophy [102]. CMR studies also reveal that HCM patients tend to have higher LV mass-to-volume ratios than athletes. In uncertain cases, particularly within the grey zone of hypertrophy, a period of detraining followed by repeat imaging can aid in diagnosis. Physiological hypertrophy typically regresses with detraining, especially wall thickness, while structural changes in HCM persist. However, some components of eccentric remodelling, such as chamber dilation, may not completely normalize, further complicating the diagnostic process.

## 9. Conclusions

HCM is a genetically and phenotypically heterogeneous disease associated with significant morbidity and an elevated risk of SCD. Accurate and early diagnosis is essential to improving patient outcomes, and cardiac imaging plays a central role throughout the clinical course of the disease. Echocardiography is the primary modality for diagnosing HCM, evaluating left ventricular hypertrophy, detecting SAM, assessing outflow tract gradients, and characterizing mitral valve anomalies. Cardiac magnetic resonance imaging provides complementary value, particularly in quantifying fibrosis, characterizing tissue, and evaluating complex morphologies. Together, echocardiography and CMR enable a multimodal, patient-centered approach that supports precise phenotyping, informs therapeutic decisions, and refines risk assessment, especially in borderline or genotype-positive/phenotype-negative cases. Imaging is also critical in distinguishing HCM from physiological adaptations in athletes and in guiding safe sports participation based on individual risk profiles. Future directions include integrating imaging biomarkers into prediction models, using artificial intelligence for automated pattern recognition, and developing personalized medicine strategies based on imaging phenotypes. Continued advances in imaging will undoubtedly enhance our understanding of and ability to manage HCM, offering new opportunities for timely intervention and improved long-term outcomes.

## Figures and Tables

**Figure 4 biomedicines-13-02138-f004:**
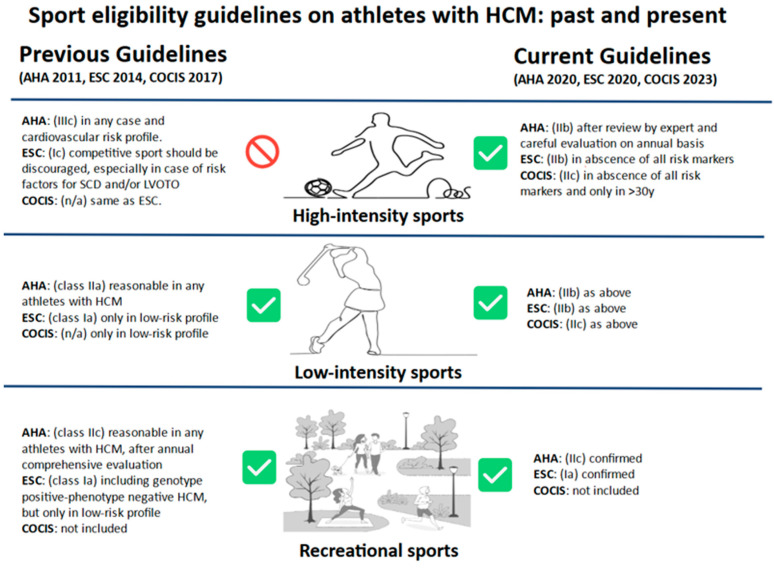
Sports eligibility guidelines about athletes with HCM: past and present. Intensity classification is different between the AHA, ESC and COCIS, but is considerably similar for the purpose of sports eligibility. Recreational exercise is performed for the purpose of leisure with no requirement for systematic training and without the purpose of excelling or competing against others. Competitive sports involve systematic training for the primary purpose of competition against others, at multiple levels, including high school, collegiate, master’s level, semiprofessional, or professional sporting activities. AHA = American Heart Association; ESC = European Society of Cardiology; COCIS = Italian Organizing Committee for Cardiac Assessment in Sports; SCD = sudden cardiac death; LVOTO = left ventricular outflow tract obstruction; HCM = hypertrophic cardiomyopathy.

**Table 1 biomedicines-13-02138-t001:** Comparative Overview of Imaging Modalities in the Evaluation of Hypertrophic Cardiomyopathy.

Modality	Main Advantages	Limitations	Clinical Role
Conventional Echocardiography (2D + Doppler)	- Widely available - First-line examination - Real-time dynamic assessment	- Operator-dependent - Limited apical visualization - Overestimates EF	Initial diagnosis, follow-up, family screening
Strain Imaging (2D-STE)	- Detects subclinical dysfunction - Correlates with fibrosis - Risk stratification	- Acoustic window limitations - Lack of standardization across vendors	Early dysfunction detection; prognosis
3D Strain Imaging/CMR Feature Tracking	- Comprehensive deformation assessment - Applicable to RV and LA - High reproducibility	- Requires advanced software - CMR not always available	Advanced phenotyping, risk stratification
Cardiac MRI (cine + LGE)	- Gold standard for volumes and mass - Fibrosis evaluation (LGE) - Complex phenotype assessment	- Higher cost - Contraindicated in some device carriers	Accurate diagnosis, prognosis, ICD indication
T1/T2 Mapping—CMR	- Assesses diffuse fibrosis - Differentiates etiologies (e.g., Fabry, amyloidosis) - Sensitive without LGE	- Technically demanding - Longer acquisition times	Early phenotyping, therapy monitoring
4D Flow CMR	- Advanced flow and turbulence analysis - Estimates energy loss - Potential predictive value	- Limited availability - Complex post-processing	Research, functional stratification
Cardiac CT	- Excellent spatial resolution - Coronary and valvular assessment - CMR alternative	- Ionizing radiation - Iodinated contrast required	CAD exclusion, morphological evaluation
AI-based Imaging (Echo, CMR, CT)	- Automated segmentation - Accurate quantification (fibrosis, volumes) - Reduces inter-operator variability	- Still under clinical validation - Depends on proprietary software/algorithms	Diagnostic support, workflow automation

## Data Availability

The original contributions presented in this study are included in the article. Further inquiries can be directed to the corresponding author(s).

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
