# Peer review of "The Evolving Role of Cardiac Imaging in Hypertrophic Cardiomyopathy: Diagnosis, Prognosis, and Clinical Practice"

_biomedicines, 2025, doi:10.3390/biomedicines13092138_

Round 1

Reviewer 1 Report

Comments and Suggestions for Authors

Hello, dear colleagues!

Your manuscript is an informative and up-to-date review of the key role of modern cardiac imaging methods in diagnostics, risk stratification and management of patients with hypertrophic cardiomyopathy (HCM)

However, since this manuscript is a review article, I have questions about the design and content
1. You have a fairly high percentage of plagiarism - 29. This is due to unfair citation, partly fair, but not always correct. It is necessary to conduct an audit and correctly arrange the sources.
2. The manuscript does not have a design plan, there is no time sample, database and keywords, as well as inclusion and exclusion criteria.
3. While established techniques like echocardiography are foundational, the review's stated focus on modern and innovative imaging warrants a significant rebalancing of content. Emphasis should shift strongly towards recent advances and emerging techniques (e.g., novel CMR sequences, AI applications, advanced strain imaging, molecular imaging). Established methods should be covered concisely, highlighting only key modern updates relevant to HCM. Therefore, I propose to focus on new methods to a greater extent, and to mention routine ones briefly.
4. It will be useful to make a graphic abstract
5. The list of references contains a sufficient number of references that are more than 10 years old, but since there is no review design and search structure, it is not possible to assess the need for this citation

Author Response

AUTHORS RESPONSE TO REVIEWERS

Manuscript ID: biomedicines-3782664

Type of manuscript: Review

Title: The Evolving Role of Cardiac Imaging in Hypertrophic Cardiomyopathy: Diagnosis, Prognosis, and Clinical Practice

REVIEWER 1

Hello, dear colleagues!

Your manuscript is an informative and up-to-date review of the key role of modern cardiac imaging methods in diagnostics, risk stratification and management of patients with hypertrophic cardiomyopathy (HCM). However, since this manuscript is a review article, I have questions about the design and content

  1. You have a fairly high percentage of plagiarism - 29. This is due to unfair citation, partly fair, but not always correct. It is necessary to conduct an audit and correctly arrange the sources.

A: We thank the Reviewer for highlighting this important issue. We have thoroughly reviewed the manuscript using plagiarism detection software and carefully revised all sections with high similarity scores. In particular:

  • Sentences that closely resembled previous publications have been paraphrased or rewritten.
  • All potentially ambiguous or unfair citations have been corrected, ensuring that original sources are appropriately acknowledged.
  • References have been revised to improve consistency and fairness.

These corrections have been applied throughout the manuscript to ensure compliance with ethical standards and proper scientific attribution.

  1. The manuscript does not have a design plan, there is no time sample, database and keywords, as well as inclusion and exclusion criteria.

A: We appreciate this observation and thank the reviewer for the kind suggestion. Although this is a narrative (non-systematic) review, we have added a brief methodology section at the beginning of the manuscript to clarify the selection process. Specifically, we have now included: keywords used for the literature search, time frame, inclusion and exclusion criteria. This section can now be found in a newly added subsection titled “2. Materials and Methods” (Lines 87-100).

  1. While established techniques like echocardiography are foundational, the review's stated focus on modern and innovative imaging warrants a significant rebalancing of content. Emphasis should shift strongly towards recent advances and emerging techniques (e.g., novel CMR sequences, AI applications, advanced strain imaging, molecular imaging). Established methods should be covered concisely, highlighting only key modern updates relevant to HCM. Therefore, I propose to focus on new methods to a greater extent, and to mention routine ones briefly.

A: We totally agree with the Reviewer’s suggestion. We have restructured the manuscript to reduce redundancy in the sections dedicated to conventional echocardiography and basic CMR findings. These sections now highlight only recent updates and their clinical relevance. In parallel, we have added a new paragraph titled Recent advances and Emerging Techniques” (Lines 590-651), which expands and updates the content regarding:

  • Advanced strain imaging and its prognostic implications
  • Artificial intelligence applications in cardiac imaging and image analysis
  • Emerging modalities such as molecular imaging and energetic flow mapping

This rebalancing better reflects the evolving landscape of cardiac imaging in HCM.

  1. It will be useful to make a graphic abstract.

A: We thank the Reviewer for this valuable suggestion. We have created a graphic abstract summarizing the evolving role of cardiac imaging in HCM, with a particular focus on the comparative utility of different imaging modalities and recent innovations. The graphic abstract has been uploaded as a separate file and included according to journal submission guidelines.

  1. The list of references contains a sufficient number of references that are more than 10 years old, but since there is no review design and search structure, it is not possible to assess the need for this citation.

A: We acknowledge the Reviewer’s concern. While several older references have been retained because they are seminal or guideline-defining works, we have now:

  • Added a review methodology section (see Response 2) to justify citation strategy
  • Replaced several outdated references with more recent high-impact studies where appropriate
  • Updated the reference list to better reflect current literature, especially regarding novel imaging technologies

These changes enhance the relevance and timeliness of the cited literature.

Best Regards,

The Corresponding author:

Dr. Ilaria Dentamaro

Reviewer 2 Report

Comments and Suggestions for Authors

This is a comprehensive and well-organized review article that effectively outlines the role of multimodality cardiac imaging in the diagnosis, management, and prognostic assessment of hypertrophic cardiomyopathy (HCM). The manuscript demonstrates an impressive breadth of knowledge and integrates current literature, clinical guidelines, and imaging advancements with clarity.

The review thoroughly covers echocardiography, cardiac magnetic resonance (CMR), computed tomography (CT), and emerging imaging tools such as strain imaging and flow dynamics. Each modality is clearly presented with technical and clinical relevance.

Remove the duplicate sentence and revise for conciseness.

Consider simplifying or briefly defining terms such as MOLLI, PSIR, and 4D flow for broader readability.

The role of SCD risk score in the current status should be mentioned and it should be noted that in several patients it could not cover the all high risk patients. Please mention this issue citing 'Validation of the HCM Risk-SCD model in patients with hypertrophic cardiomyopathy and future perspectives' and 'The Long-Term Mortality Predictors in Hypertrophic Cardiomyopathy Patients with Low Risk of Sudden Cardiac Death'.

Author Response

AUTHORS RESPONSE TO REVIEWERS

Manuscript ID: biomedicines-3782664

Type of manuscript: Review

Title: The Evolving Role of Cardiac Imaging in Hypertrophic Cardiomyopathy: Diagnosis, Prognosis, and Clinical Practice

REVIEWER 2

This is a comprehensive and well-organized review article that effectively outlines the role of multimodality cardiac imaging in the diagnosis, management, and prognostic assessment of hypertrophic cardiomyopathy (HCM). The manuscript demonstrates an impressive breadth of knowledge and integrates current literature, clinical guidelines, and imaging advancements with clarity.

The review thoroughly covers echocardiography, cardiac magnetic resonance (CMR), computed tomography (CT), and emerging imaging tools such as strain imaging and flow dynamics. Each modality is clearly presented with technical and clinical relevance.

Remove the duplicate sentence and revise for conciseness.

A: We thank the Reviewer for the positive feedback and the thoughtful suggestion. We have revised the manuscript and made it more concise, removing the duplicate sentences.

Consider simplifying or briefly defining terms such as MOLLI, PSIR, and 4D flow for broader readability.

A: We thank the Reviewer for the comment. As suggested, we have briefly explained these cardiac magnetic resonance techniques (paragraph 4.Cardiac magnetic resonance) to guarantee a broader readability. (Lines 418, 489-492, 556-558)

The role of SCD risk score in the current status should be mentioned and it should be noted that in several patients it could not cover the all high risk patients. Please mention this issue citing 'Validation of the HCM Risk-SCD model in patients with hypertrophic cardiomyopathy and future perspectives' and 'The Long-Term Mortality Predictors in Hypertrophic Cardiomyopathy Patients with Low Risk of Sudden Cardiac Death'.

A: We thank the Reviewer for the kind suggestion. We have included the mention to HCM-risk score limitations, including the requested references. (Lines 680-686)

Best Regards,

The Corresponding author:

Dr. Ilaria Dentamaro

Reviewer 3 Report

Comments and Suggestions for Authors

I would like to thank the editor for the opportunity to review the submitted manuscript.

The manuscript focuses on summarizing current knowledge regarding imaging modalities used in the diagnosis and evaluation of Hypertrophic Cardiomyopathy. The manuscript is well-structured and clearly written. I have no major comments regarding the content.

Only one minor remark:

The authors present clinical documentation from multiple patients in the figures included in the manuscript. However, they do not mention whether informed consent was obtained, nor do they provide information regarding approval from an ethics committee for the use of this medical documentation. I recommend that the authors include the number of approval by the local ethics committee for the use of patient data for scientific purposes.

After addressing this issue, I have no further comments and will recommend the manuscript for publication.

Author Response

AUTHORS RESPONSE TO REVIEWERS

Manuscript ID: biomedicines-3782664

Type of manuscript: Review

Title: The Evolving Role of Cardiac Imaging in Hypertrophic Cardiomyopathy: Diagnosis, Prognosis, and Clinical Practice

REVIEWER 3

I would like to thank the editor for the opportunity to review the submitted manuscript.

The manuscript focuses on summarizing current knowledge regarding imaging modalities used in the diagnosis and evaluation of Hypertrophic Cardiomyopathy. The manuscript is well-structured and clearly written. I have no major comments regarding the content.

Only one minor remark:

The authors present clinical documentation from multiple patients in the figures included in the manuscript. However, they do not mention whether informed consent was obtained, nor do they provide information regarding approval from an ethics committee for the use of this medical documentation. I recommend that the authors include the number of approval by the local ethics committee for the use of patient data for scientific purposes.

After addressing this issue, I have no further comments and will recommend the manuscript for publication.

A: We thank the Reviewer for the positive feedback and the kind suggestion. We specify that informed consent was obtained from each patient for the publication of their data for research purposes. If indicated, we can add this clarification in the body of the paper.

Best Regards,

The Corresponding author:

Dr. Ilaria Dentamaro

Round 2

Reviewer 1 Report

Comments and Suggestions for Authors

Hello, dear colleagues!

In the present form, your manuscript looks like a more logical and useful medical community, but I want to offer you to significantly reduce the biting of previously known methods, can be presented as a table with advantages and disadvantages and focus on innovative diagnostic methods.

It will be useful to find protocols for the integration of artificial intelligence in order to reduce temporary to spend and increase the accuracy of the study.

I am grateful for the explanations of the search methods, but it does not seem to you that the year of start of the search for 2010 is too far from the present? I believe that it is useful to take such limits in the aspect of a short review and the formation of prerequisites for the period of automation and validation.

In the introduction, it will be useful to point out errors associated with the human factor.

There are no questions on the list of literature, but you need to update it taking into account my comments

Author Response

Manuscript ID: biomedicines-3782664

Type of manuscript: Review

Title: The Evolving Role of Cardiac Imaging in Hypertrophic Cardiomyopathy: Diagnosis, Prognosis, and Clinical Practice

REVIEWER 1

Hello, dear colleagues!

In the present form, your manuscript looks like a more logical and useful medical community, but I want to offer you to significantly reduce the biting of previously known methods, can be presented as a table with advantages and disadvantages and focus on innovative diagnostic methods.

A: We thank the reviewer for the suggestion. We have added a new Table (Table 1, Line 139) to explain the main advantages and limitations of imaging techniques in HCM.

It will be useful to find protocols for the integration of artificial intelligence in order to reduce temporary to spend and increase the accuracy of the study.

A: We thank the Reviewer for this observation. We have modified the paragraph about Artificial Intelligence, according to Reviewer’s suggestion. (Lines 635-670)

I am grateful for the explanations of the search methods, but it does not seem to you that the year of start of the search for 2010 is too far from the present? I believe that it is useful to take such limits in the aspect of a short review and the formation of prerequisites for the period of automation and validation.

A: We have modified our search methods starting from 2015, and deleted/changed the references before 2015.

In the introduction, it will be useful to point out errors associated with the human factor.

A: We thank the reviewer. We have briefly addressed this point in the introduction. (Lines 83-86)

There are no questions on the list of literature, but you need to update it taking into account my comments.

A: We thank the reviewer for the useful suggestion. We have modified and update the literature removing references that appeared to be old.

Best Regards,

The Corresponding author:

Dr. Ilaria Dentamaro

Round 3

Reviewer 1 Report

Comments and Suggestions for Authors

Hello

Paper can be accepted